# A Promising Approach for Primary Cytoreductive Surgery for Advanced Ovarian Cancer: Survival Outcomes and Step-by-Step Description of Total Retroperitoneal en-Bloc Resection of Multivisceral-Peritoneal Packet (TROMP)

**DOI:** 10.3390/jpm12060899

**Published:** 2022-05-29

**Authors:** Mustafa Zelal Muallem, Luisa Kluge, Ahmad Sayasneh, Jalid Sehouli, Dario Zocholl, Jumana Muallem, Andrea Miranda

**Affiliations:** 1Department of Gynecology with Center for Oncological Surgery, Charité—Universitätsmedizin Berlin, Corporate Member of Freie Universität Berlin, Humboldt-Universität zu Berlin and Berlin Institute of Health, Virchow Campus Clinic, Charité Medical University, 13353 Berlin, Germany; luisa.kluge@charite.de (L.K.); jalid.sehouli@charite.de (J.S.); dario.zocholl@charite.de (D.Z.); jumana.muallem@charite.de (J.M.); andrea.miranda@charite.de (A.M.); 2Department of Gynecological Oncology, Surgical Oncology Directorate, Guy’s and St Thomas’ NHS Foundation Trust, School of Life Course Sciences, Faculty of Life Sciences and Medicine, King’s College London, Westminster Bridge Road, London SE1 7EH, UK; ahmad.sayasneh@gstt.nhs.uk

**Keywords:** advanced ovarian cancer, total retroperitoneal en-bloc resection of multivisceral-peritoneal packet, TROMP, primary cytoreductive surgery, complete tumor resection, survival outcomes

## Abstract

(1) Background: A complete tumor resection during primary cytoreductive surgery has been reported to be the most important and perhaps the only independent prognostic factor in advanced ovarian cancers. The goal of complete cytoreduction needs to be weighed against the potential morbidities and long-term survival outcomes. (2) Methods: in this retrospective analysis of a prospectively obtained database, 208 consecutive patients with advanced ovarian cancer who underwent a conventional primary cytoreductive surgery (150 patients) or TROMP technique (58 patients) were included. Progression-free and overall survival rates were calculated using Kaplan–Meier analysis as well as the 95% confidence interval of the hazard ratio between treatment groups. (3) Results: After a median follow-up phase of more than 3 years (range 1–72 months), there are no statistically significant differences between both groups in progression-free and overall survival rates. Albeit, the TROMP group included statistically significant more advanced-stage cases compared to the conventional surgery group. (4) Conclusions: the TROMP technique is a promising tool for successful primary cytoreductive surgery in a selected group of patients with high tumor burdens in order to achieve optimal surgical results and survival outcomes without introducing any additional risks or complications.

## 1. Introduction

When it comes to ovarian cancer survival rates, Europe as a whole is rather diverse. Mortality rates ranged from 7.6 instances per 100,000 people in Portugal to 18.7 cases per 100,000 people in Latvia, with 12.9 cases per 100,000 women in Germany slightly above the average [1]. Ovarian cancer, nevertheless, continues to have the lowest survival rate of all gynecological cancers. When the disease is in its early stage, the mortality rate is much lower than it is as the disease progresses. According to the classification of the International Federation of Gynecology and Obstetrics (FIGO) [2], in the early stages (FIGO I), there is a relatively high 5-year survival rate of 80–85%, whereas patients diagnosed with very advanced stages (FIGO IIIC and FIGO IV) have a survival rate of just 25% or less [3]. However, when diagnosed, 75% of patients had advanced epithelial ovarian cancer (FIGO III and FIGO IV), explaining the dismal median 5-year survival rate [4]. Progression-free and overall survival rates can be predicted by a complete tumor resection during the first line cytoreduction, which has been shown to be the most important and perhaps the only independent prognostic factor [5,6,7]. The rate of complete tumor resection, on the other hand, is heavily dependent on the experience, available resources, and infrastructures of the sites where primary cytoreductive surgery is performed. Patients who underwent extended surgical resection had a significantly higher 5-year disease-specific survival rate than those who underwent fewer radical procedures (44% vs. 17%, *p* < 0.001), according to a study that examined the impact of radical surgical procedures on the survival of ovarian cancer patients [8]. Over the last two decades, many surgical groups gradually registered increased rates of complete tumor resection of advanced ovarian cancer [9,10,11,12,13]. In 2020 [14], we introduced a surgical technique for advanced ovarian cancer therapy that is both practical and highly effective. This technique is called “total retroperitoneal en bloc resection of multivisceral-peritoneal packet” (TROMP operation). TROMP has enhanced the complete tumor resection rate to 87.9% in advanced ovarian patients without increasing blood loss, postoperative complications, or surgical time. The procedure is a no-touch isolation technique performed in the retroperitoneal area in order to resect the parietal peritoneum and afflicted organs of advanced ovarian cancer. When we introduced TROMP as a novel surgical modality, we were extremely cautious to ensure that it did not affect cancer-related outcomes. As a result, we sought to publish the survival data from the aforementioned trial and to detail the TROMP technique for primary cytoreductive surgery in advanced ovarian patients in this article.

## 2. Materials and Methods

This is a retrospective analysis of a prospectively obtained database. The study was approved by our institutional review board (under registration number EK207/2003 Amendment 15/2012). The study included patients who were referred to our Department of Gynecology, Center of Oncological Surgery at the Charité Medical University of Berlin between January 2015 and December 2017. Clinical data on consecutive patients with primary epithelial advanced ovarian cancer (FIGO stage III-IV) were collected prospectively from the ovarian cancer tumor bank in our database (www.toc-network.de) (last accessed on 12 May 2022). Patients with non-epithelial ovarian cancer or borderline tumors were excluded from the study, as were those who had interval cytoreductive surgery or only a second look operation or diagnostic procedure. Similarly, patients with early-stage epithelial ovarian cancer (FIGO stages I and II) were excluded as well. Total retroperitoneal en-bloc resection of multivisceral-peritoneal packet (TROMP operation) was developed in 2013 at our institution by the first author (M.Z.M). The learning curve extended from 2013 to 2015 and included 25 patients [14]. The TROMP technique also featured a nerve-sparing systematic lymph node dissection as part of the procedure [15,16] wherever it was indicated. Informed consent was obtained from all patients prior to clinical data collection. The postoperative complications in this study were graded according to the Calvien-Dindo classification [17] and the perioperative morbidity and mortality were defined as any adverse event occurring within 30 days of surgery. These data were published in our first paper [13]. The adjuvant treatments were indicated and performed as per routine clinical practice and patient preferences. The term optimal cytoreduction was used for cases with no macroscopic residual disease [18].

Statistical analyses were performed using IBM SPSS Statistics 21.0 (SPSS, Chicago, IL, USA). Frequency counts and percentages were used to describe categorical variables, and continuous variables were summarized as the median and range. Statistical significance was defined by a *p* < 0.05 and 2-sided tests were applied. Overall survival was calculated from the day of primary cytoreductive surgery until the day of death from any cause (event) or the last day of follow-up (censored). Losses to follow-up were regarded as censored observations. Progression-free survival was calculated from the day of cytoreductive surgery until the day of diagnosis of the first relapse, the day of death from any cause or the last day of follow-up, which happens earlier. Progression-free and overall survival were calculated using Kaplan–Meier analysis as well as the 95% confidence interval of the hazard ratio between treatment groups. To evaluate the significance of differences in survival, the *p*-values of the corresponding coefficients of univariable and multivariable Cox regression analysis were used.

### 2.1. Surgical Steps of TROMP Technique

#### 2.1.1. The Incision

1.Incising the abdomen in the midline from the pubic symphysis to the umbilicus if the diagnosis is not already histologically confirmed (diagnostic laparoscopy or biopsy). If the diagnosis is already histologically confirmed, the incision will be extended to the xiphoid process.2.Dissecting the fat tissue from the rectus fascia (rectus sheath) for about 1 cm lateral from the middle line (linea alba). This maneuver facilitates retracting the abdominal wall and closing of the rectus fascia (Figure 1).

3.Incising the rectus fascia at the midline, assuring not to injure the underlying parietal peritoneum (Figure 2).

4.For the histologically unconfirmed diagnosis, open a peritoneal window in the midline for approximately 3 to 4 cm to take the biopsies and to evaluate the peritoneal cavity. In such cases, we evaluate the infiltration of the small bowel, especially the serosa and the infiltration of the hepatoduodenal ligament. These two locations may be a reason for not achieving a complete tumor resection.

#### 2.1.2. Parietal Peritonectomy in the Upper Abdomen

5.For cases with a confirmed diagnosis of epithelial carcinoma, we keep the parietal peritoneum intact and dissect it using a curved bipolar Metzenbaum scissors (G. F. Mersons Limited, Ethicon Suture Laboratories, Bridgewater, New Jersey, USA) and Mikulicz clamps on the incision edges of the rectus fascia in order to provide traction along the extent of the tissue transection plane (Figure 3).

6.After dissecting the parietal peritoneum to the lateral side of the renal capsule bilaterally, triangular ligament of the liver on the right side and on the upper edge of the spleen and to the round ligament at the pelvic inlet bilaterally, the abdominal retractor system could now be used to achieve broad access to the posterior parietal peritoneum, especially at the diaphragm. We use the Sattler ^®^ Königsee retractor system (Medizintechnik Sattler GmbH, Königsee-Rottenbach, Germany) (Figure 4).

7.Developing the dissection in the upper abdomen first at the midline beneath the xiphoid process outlining the diaphragmatic central tendineum moving from the right side behind the coronary ligament of the liver to the left, exposing the diaphragmatic muscle and the left adrenal gland. The dissection must be performed carefully at the midline and to the right to expose the vena cava 3 to 4 cm beneath the xiphoid process (Figure 5).

8.By following the right side of the vena cava, the right inferior phrenic veins can be identified and spared. To enter the bare portion of the liver, one should strip the peritoneum from the right border of the centrum tendineum and dissect the right triangular ligament of the liver. This dissection must be finished medially and beneath the liver in order to completely separate it from the Gerota fascia, adrenal gland, and renal capsule (Figure 6).

9.In the middle of the coronary ligament, one should detach the peritoneal reflection from the liver caudally to the hepatic falciform ligament insertion in the liver, which will be cut and ligated. The liver is now completely mobilized and free from the peritoneum (Figure 7a,b).

10.Dissecting the peritoneum from the renal capsule and exposing the course of the vena cava may be performed easily and one should proceed to the hepatoduodenal ligament, then from the laterocranial side to the mediocaudal side of the duodenum (Figure 8).

#### 2.1.3. Omentectomy/Splenectomy

11.Resection of the greater omentum should be performed starting from the greater curvature of the stomach to open the bursa omentalis completely (Figure 9).

12.If there is a need to perform a splenectomy, the resection of the short gastric vessels should be performed toward the dissection plane of the parietal peritoneum at the bottom of the left diaphragm. The parietal peritoneum will stay here, connected to the gastrocolic and gastrosplenic ligament.13.Proceeding with the peritonectomy around the spleen and stripping the peritoneum from the left adrenal gland, the tail of the pancreas and the left renal capsule should expose the splenic artery and vein. These will be cut and ligated from the retroperitoneal side under visual control of the pancreatic tail (Figure 10). This peritoneal resection will be continued here at the lower edge of the pancreas to the renal hilus.

#### 2.1.4. Peritonecomy in Mid-Abdomen and Preparing the Retroperitoneal Space

14.The omental cake is then resected from the transverse colon. The resection is performed from the white line of Toldt along the lateral aspect of the ascending and descending colon to the pelvic inlet. (Figure 11). This will allow the packing of the entire parietal peritoneum/omentum packet in a surgical towel to place it caudally out of the abdomen.

15.The dissection will be developed medially from the ascending colon to the radix mesentrii. With this manoeuvre, it is possible to pack the bowel in a surgical towel and mobilize it cranially outside the abdomen and expose the entire retroperitoneal space and the main vessels.16.Identification of the ovarian vessels and ureter bilaterally should be performed and the ovarian vessels should be transected at the junction of the inferior cava and left renal vein (Figure 12).

17.Preparing the superior hypogastric plexus and sparing it in the midline of the retroperitoneal space directly above the aortic bifurcation is essential (Figure 13).

#### 2.1.5. Pelvic Peritonectomy with Resection of the Pelvic Packet

18.The round ligament should be transected retroperitoneally and the medial umbilical ligament should be identified and highlighted caudally at the level of the uterine artery to recognize the lateral and inferior bladder walls.19.Dissecting the bladder from the uterus and cervix above the level of the uterine artery from the medial umbilical ligament is performed next. At this level, the bladder is not covered with peritoneum and it will be dissected easily by pushing it ventrally (Figure 14). Stripping the peritoneum from the dome of the bladder is then performed from the bottom up.

20.When the bladder is completely stripped from the peritoneum, the peritoneal packet is moved to the later aspect outside the abdomen in order to expose the anterior and lateral sides of the pelvic packet (Figure 15).

21.The uterine vessels are now easy to identify, cut and ligated, as well as the lateral parametrium (Figure 16). Care should be taken to spare the hypogastric nerve and ureter.

22.The anterior vaginal wall should be opened (Figure 17a), cutting and ligating the lateral vaginal wall and opening the posterior vaginal wall without incising the Douglas peritoneum. Sacrouterine ligaments are resected and ligated retroperitoneally (Figure 17b). Again, care has to be taken to not injure the hypogastric nerves.

23.Next, one should dissect the rectosigmoid colon from the posterior vaginal wall, directly beneath the lowest point of the pouch of Douglas. If there is no indication to resect a rectosigmoid, the Douglas pouch peritoneum should be dissected and the whole packet removed (Figure 18).

24.If the resection of the rectosigmoid is indicated, the blood supply should be centrally ligated to assure a no-touch isolation. The resection of the mesorectum is performed above the level of the hypogastric nerve and the rectosigmoid is resected with a stapler (Figure 19). The entire multivisceral-peritoneal packet should be removed and the vaginal vault closed. A circular stapling device is used to complete the colorectal anastomosis (Intraluminal Stapler 29, Ethicon, Cincinnati, OH) and the multivisceral peritoneal packet is removed as one specimen (Figure 20).

25.One should then resect any residual tumor from the intestinal mesenterium (Figure 21), omentum minus or hepatic capsule. Performing any other indicated cytoreductive procedures to achieve the complete tumor resection is recommended.

26.If radical lymphadenectomy is indicated, it should be performed sparing the superior hypogastric plexus, the aortic plexus, the lumbar splanchnic nerves and the mesenteric plexus (Figure 22a,b). The operation is available in full length as Appendix A.

## 3. Results

A total of 208 patients who met the inclusion criteria were included in the study; 150 patients with primary advanced ovarian cancers were operated on with the conventional surgical method (opening the intraperitoneal cavity, exploring the tumor dissemination pattern, resecting and deperitonealising the affected organs and areas separately using the intraperitoneal dissections planes) and 58 patients were operated using the innovative TROMP (total retroperitoneal en-bloc resection of multivisceral-peritoneal packet) technique, which enables the surgeon to perform an en-bloc retroperitoneal resection of all the peritoneal tumor (not only the pelvic tumor) combined with the no-touch technique by retroperitoneal resection of the affected organs.

A detailed description of the study population and the setting (patients and surgery characteristics) has been previously reported elsewhere [14]. Only a brief summary is given here.

A total of 75.7% of all patients had intraoperative evaluated FIGO III, whereas 20.7% of all patients had FIGO IV according to the intraoperative surgical evaluation (25.9% in the TROMP group vs. 18.7% in the conventional surgery group, *p* = 0.34). The TNM classification revealed a significant difference between the groups for advanced stages (T3c+T4); 81% in the TROMP group versus 64% in the conventional surgery group (*p* = 0.03). Infiltration of the upper abdomen was found in 82.8% of the TROMP group and in 64.7% of the conventional surgery cases (*p* = 0.02). Optimal cytoreductive surgery to no visible disease was performed in 68.8% of all patients; a complete tumor resection rate was performed in 87.9% of the patients in the TROMP group and 61.3% of the patients in the conventional surgery group (*p* = 0.001). This difference was more significant in the sub-group of very advanced stages (T3c+T4). The resection to no visible disease was achieved in 85.1% of the patients in the TROMP group and in only 53.1% of the patients in the conventional surgery group (*p* = 0.001). Surgery to <10 mm residual tumor was achieved in 100% of the patients in the TROMP group and in 90.7% of the patients in the conventional surgery group (*p* < 0.05). There were no differences between the two groups in regard to rates of blood transfusions (median = 1 unit in the TROMP group vs. 2 units in the conventional surgery group, *p* = 0.58) and fresh frozen plasma concentrates (median = 10 units in the TROMP group vs. 12 units in the conventional surgery group, *p* = 0.45). The higher rate of complex procedures in the TROMP group compared with the conventional surgery group (para-aortic lymph node dissection (82.8% vs. 59.3%, *p* = 0.003), appendectomy (43.1% vs. 27.3%, *p* = 0.04), small bowel resection (32.8% vs. 15.3%, *p* = 0.009), large bowel resection (82.8% vs. 58.7%, *p* = 0.002), partial liver resection (17.2% vs. 5.3%, *p* = 0.014), and splenectomy (43.1% vs. 8%, *p* < 0.001)) was not associated with increased blood loss, the median admission time to the intensive care unit, or the rate of postoperative complications. In spite of the significantly increased rate of advanced surgical procedures in the TROMP arm, the surgical duration was approximately 33 min shorter in the TROMP group (median 335 min vs. 368 min for TROMP vs. conventional surgery, respectively) (*p* = 0.113, Mann–Whitney U Test).

The TROMP technique did not associate with increased blood loss, the median length of stay at the intensive care unit, or the rate of postoperative complications.

### Survival Outcomes

The overall population of the patients available for survival analysis is 146 patients from the conventional surgery group and 53 patients from the TROMP operation technique group. The dropout rate is 2.7% vs. 8.6% in the group of conventional surgery and the TROMP operation group, respectively. After a median follow-up phase of more than 3 years (range 1–72 months), there are no statistically significant differences between both groups regarding progression-free and overall survival.

The 3-year and 5-year progression-free survivals were 36% and 17.1% vs. 30% and 23.6% for the conventional and TROMP group, respectively. The median progression-free survival was 26.3 vs. 18.43 months (HR = 1.19, *p* = 0.395). Figure 23a shows the Kaplan–Meier curves of progression-free survival in both groups. Hazard ratio (HR) for relapse in the TROMP operation group = 0.99, *p* = 0.976 by a stratified log-rank test with stratification according to the tumor stage.

The 3-year and 5-year overall survivals were 67.8% vs. 46.72% and 49.1% vs. 47.6% for the conventional and TROMP group, respectively. The median overall survival was 59.97 vs. 46.72 months (HR = 1.36, *p* = 0.19). Figure 23b illustrates the Kaplan–Meier curves of the overall survival in both groups.

## 4. Discussion

The TROMP approach seems to be an excellent tool for increasing visibility and exposure by relocating the dissection plane from the intraperitoneal to retroperitoneal space, minimizing blood loss, and shortening the procedure’s length. In patients with highly advanced ovarian cancer, the TROMP group obtained a greater rate of full tumor excision in a shorter median surgery time [14]. After a long follow-up, the survival analysis reveals no difference between both groups regarding 3-year and 5-year progression-free and overall survival. This seems somewhat surprising, taking into account the statically significant complete tumor resection rate (completeness of cytoreduction (CC)-0 score [19]) in the TROMP group in comparison with the conventional surgery group (87.9% vs. 61.3%, *p* = 0.001). However, this is straightforward, to be explained by the statistically significant more advanced-stage cases in the TROMP group than in the conventional surgery group. This is even without taking into account the hypothesis that the advanced tumor stage (carcinomatoses) at the time of the first diagnosis of ovarian cancer corresponds to the tumor’s more aggressive biological nature [5,20,21]. This, according to those authors, makes the survival benefit associated with optimal surgery limited to patients with a less aggressive disease, while the tumor biology stays as the primary survival determinant in more advanced cases.

The median progression-free and overall survival rates in this study are very good and comparable to other studies that included patients in less advanced stages. The median progression-free and overall survivals were 18.43–26.30 months and 46.72–59.97 months in our study in comparison with the median overall survival of 45 months for stages IIB- IV ovarian cancer patients, who were operated on after the establishment of the specific ovarian cancer quality management program reported by Harter et al. [9]. Median progression-free and overall survivals were 15 and 41 months for patients assigned to primary debulking surgery in the newly published data of the SCORPION trial, where 5-year progression-free and overall survival were, respectively, 4.5% and 39.7% in this group of patients [13].

Because of the small number of patients in the TROMP group, our study may have been constrained in terms of discovering statistically significant differences in survival analysis, as well as through retrospective analysis and interpretation, which could explain the heterogeneity of the groups. Notwithstanding the above, we believe that our current study highlights the positive results of the TROMP technique, particularly in a selected group of patients with high tumor loads, in order to achieve better surgical and survival outcomes without introducing any additional risks or complications. It is still strongly recommended that a randomized controlled trial be conducted to compare the TROMP technique with conventional surgical procedures for primary cytoreductive surgery.

## 5. Conclusions

The TROMP technique is a promising tool for successful primary cytoreductive surgery in a selected group of patients with high tumor burdens in order to achieve optimal surgical results and enhance survival outcomes without increasing surgical-specific morbidity. Conducting a randomized controlled trial to compare the TROMP technique with conventional surgical procedures is strongly recommended.

## Figures and Tables

**Figure 1 jpm-12-00899-f001:**
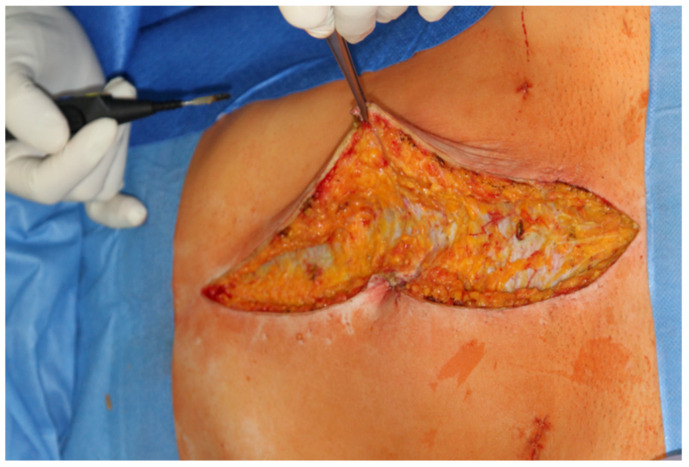
Dissecting the fat tissue from the rectus fascia (rectus sheath) for about 1 cm lateral from the middle line to facilitate the retracting of the abdominal wall.

**Figure 2 jpm-12-00899-f002:**
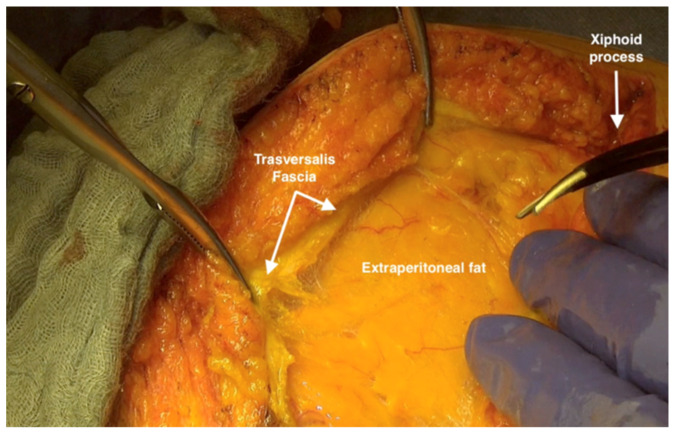
Incising the rectus fascia at the midline, assuring not to injure the underlying parietal peritoneum.

**Figure 3 jpm-12-00899-f003:**
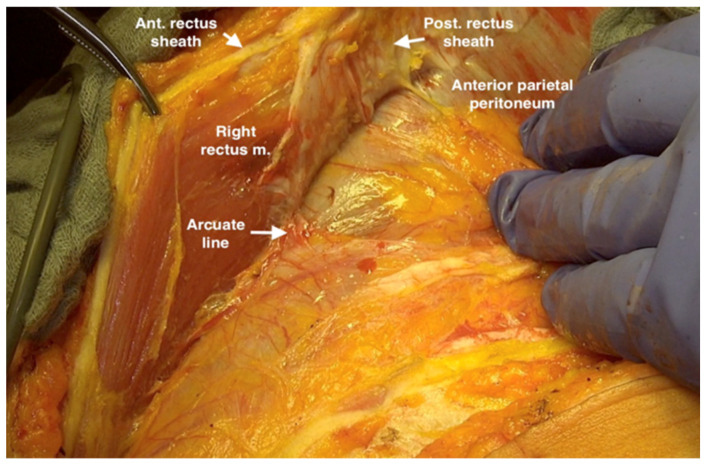
Dissecting the parietal peritoneum from the fascia at the posterior wall of the rectus abdominus.

**Figure 4 jpm-12-00899-f004:**
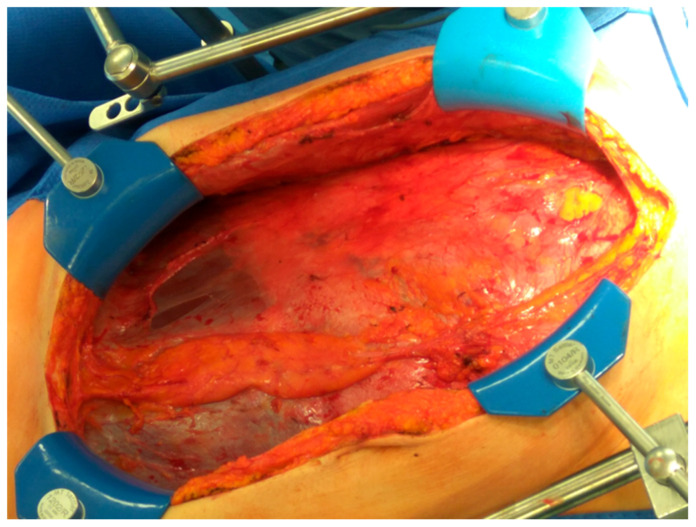
Using the Sattler Königsee retractor system for enhanced visibility and exposure.

**Figure 5 jpm-12-00899-f005:**
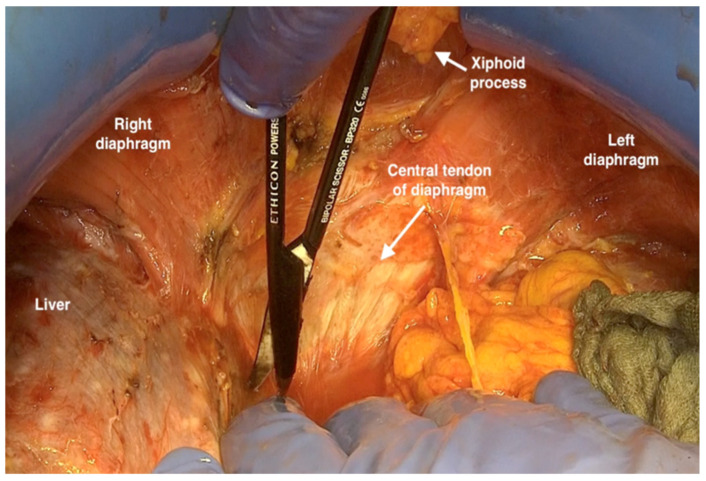
Developing the dissection in the upper abdomen with deperitonealising the centrum tendineum.

**Figure 6 jpm-12-00899-f006:**
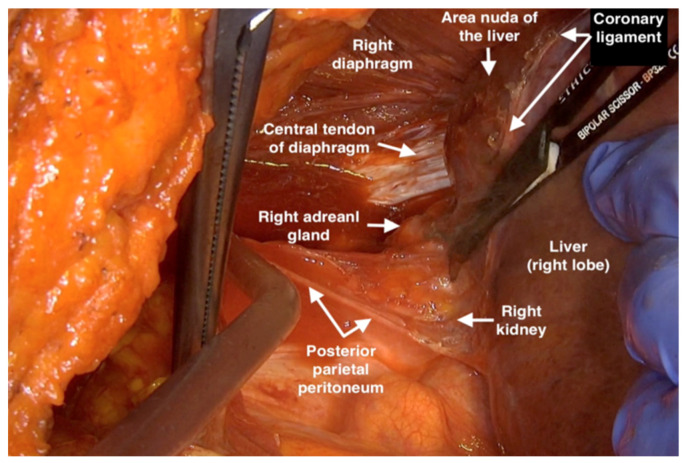
Dissecting the right triangular ligament of the liver and moving it to the middle line to expose the whole diaphragmatic area.

**Figure 7 jpm-12-00899-f007:**
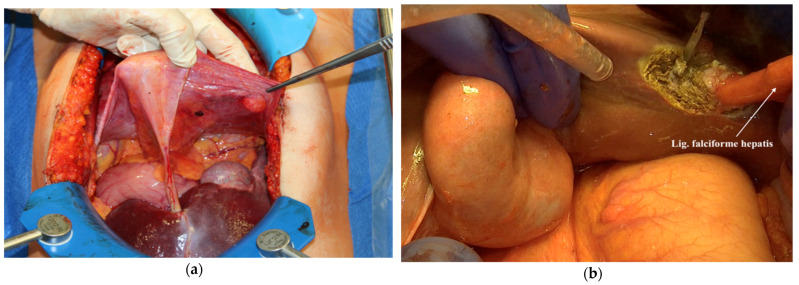
The liver is only attached to ligamentum falciforme hepatis; (**a**) Resecting the ligamentum falciforme hepatis from the liver tissue (**b**).

**Figure 8 jpm-12-00899-f008:**
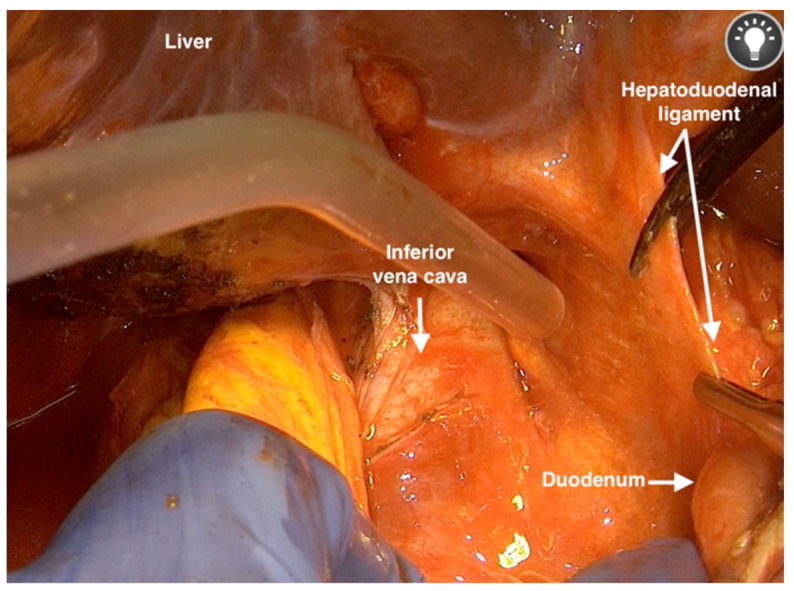
Dissecting the peritoneum from the renal capsule and exposing the course of the vena cava to the hepatoduodenal ligament.

**Figure 9 jpm-12-00899-f009:**
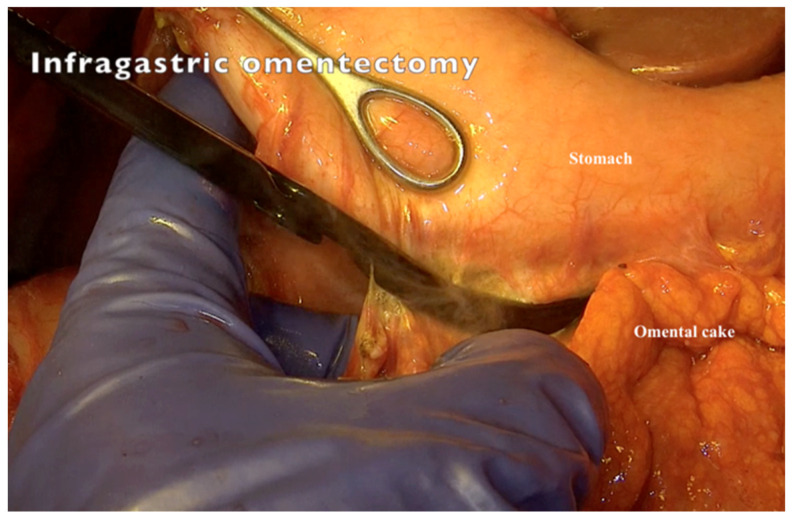
Resecting omentum majus from the greater curvature of the stomach and opening the bursa omentalis.

**Figure 10 jpm-12-00899-f010:**
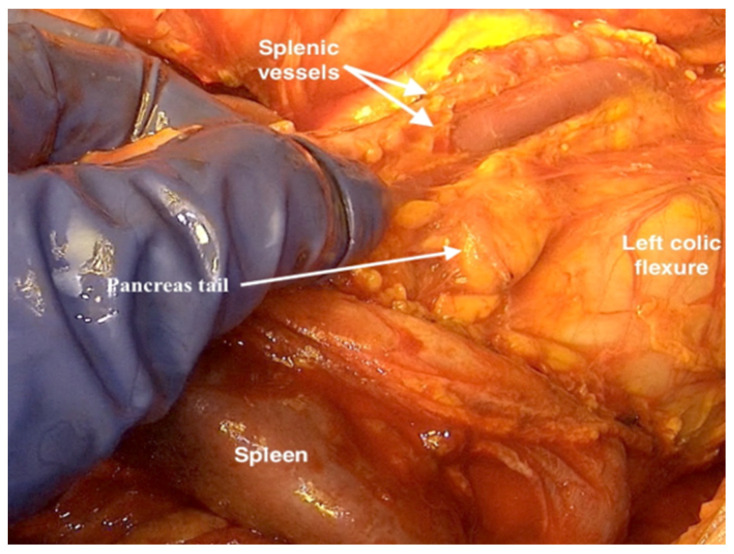
Resecting the splenic artery and vein from the retroperitoneal side under visual control of the pancreatic tail in case of tumor infiltration of the spleen.

**Figure 11 jpm-12-00899-f011:**
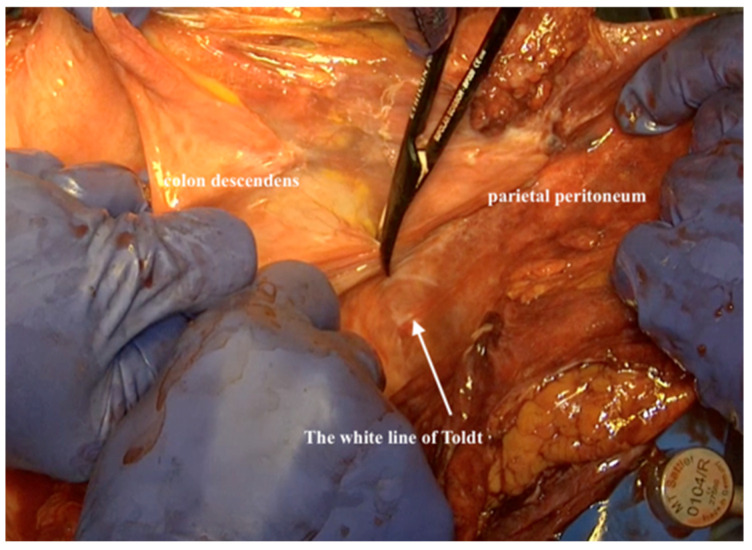
Resecting the peritoneum from the white line of Toldt along the lateral aspect of the ascending and descending colon to the pelvic inlet.

**Figure 12 jpm-12-00899-f012:**
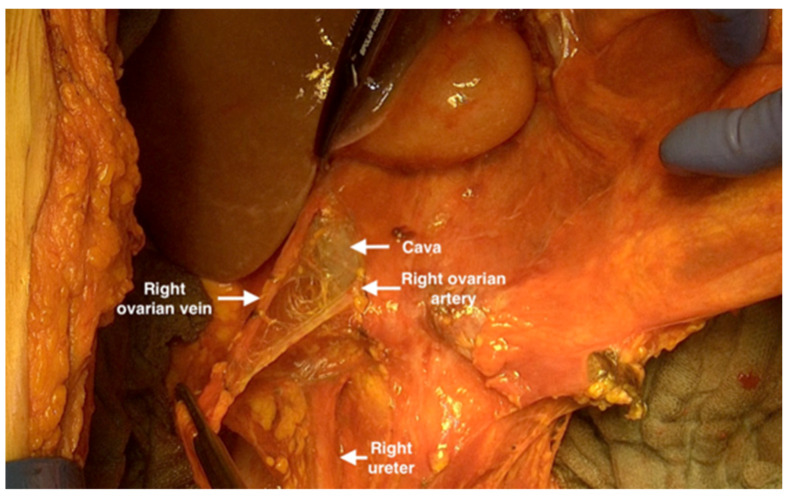
Transecting the ovarian vessels at the junction of the inferior cava.

**Figure 13 jpm-12-00899-f013:**
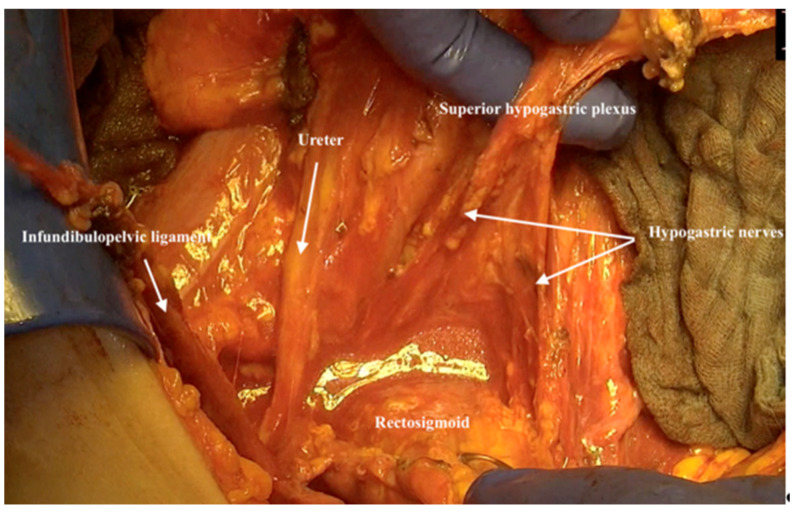
Preparing the superior hypogastric plexus and sparing it in the midline of the retroperitoneal space directly above the aortic bifurcation.

**Figure 14 jpm-12-00899-f014:**
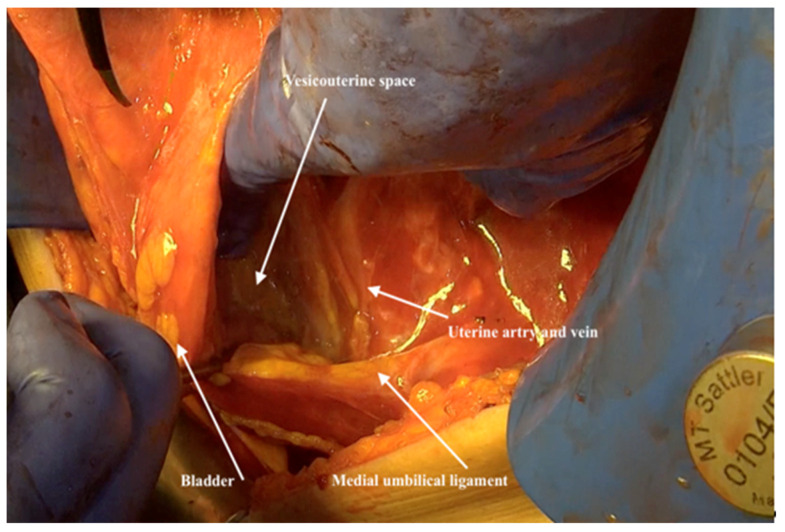
Dissecting the bladder from the uterus and cervix above the level of the uterine artery behind the medial umbilical ligament.

**Figure 15 jpm-12-00899-f015:**
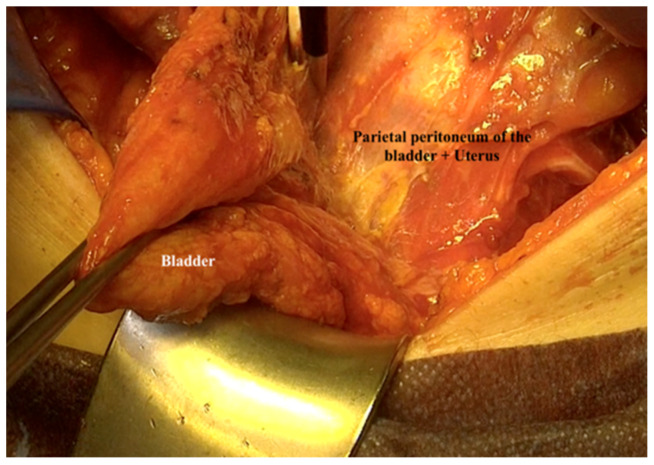
Exposing the anterior and lateral sides of the pelvic packet after stripping the bladder from the pelvic peritoneum.

**Figure 16 jpm-12-00899-f016:**
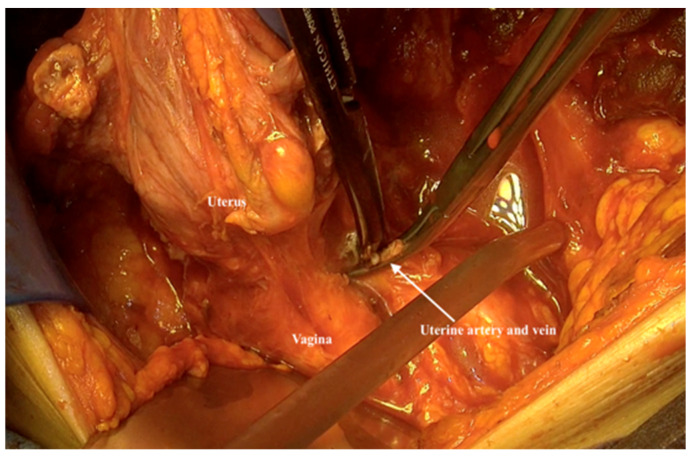
Cutting the uterine vessels and lateral parametrium without injuring the hypogastric nerves.

**Figure 17 jpm-12-00899-f017:**
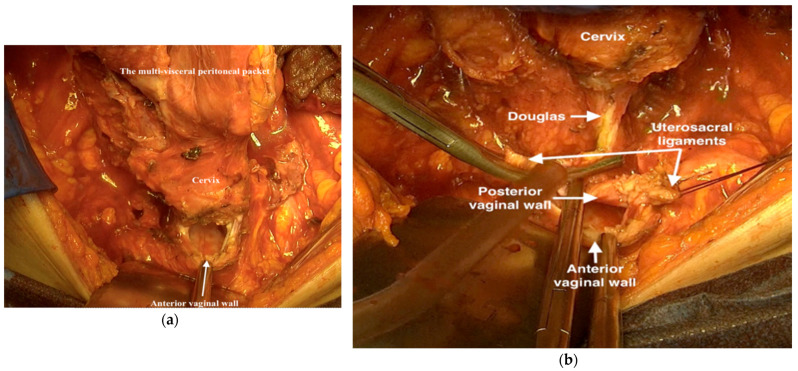
Opening the anterior vaginal wall; (**a**) cutting and ligating the lateral vaginal wall and opening the posterior vaginal wall without incising the Douglas peritoneum (**b**).

**Figure 18 jpm-12-00899-f018:**
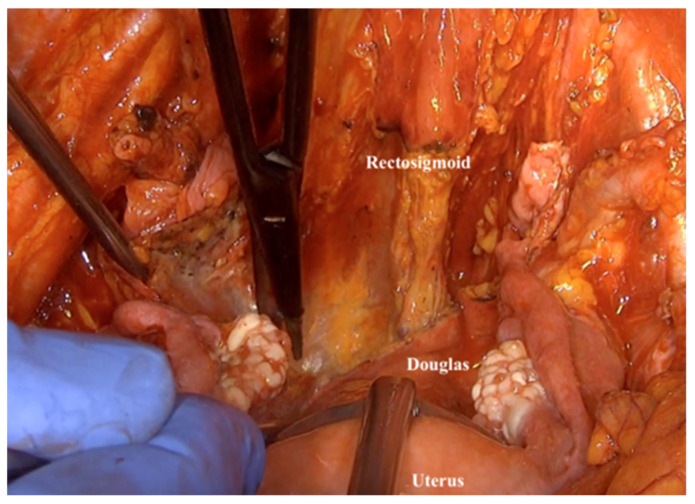
Dissecting the Douglas pouch peritoneum, when the rectosigmoid resection is not indicated.

**Figure 19 jpm-12-00899-f019:**
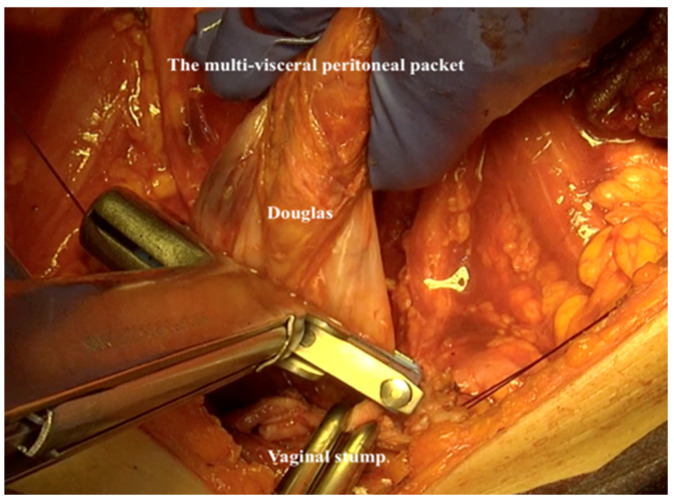
Resecting mesorectum above the level of hypogastric nerve and resecting the rectosigmoid retroperitoneally with a stapler.

**Figure 20 jpm-12-00899-f020:**
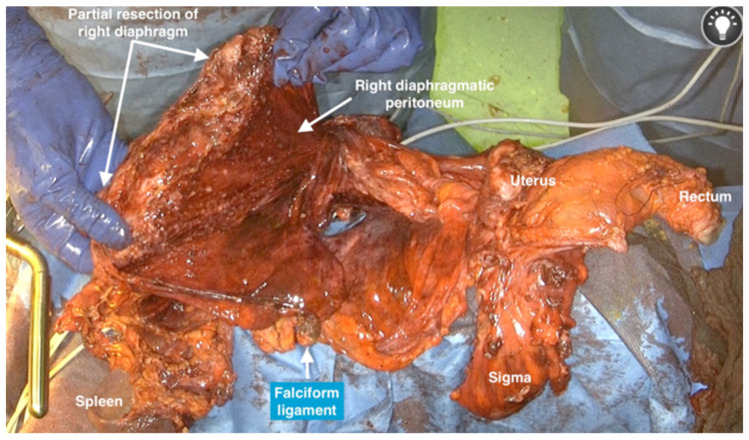
The final specimen following the use of the total retroperitoneal en-bloc resection of multivisceral-peritoneal packet (TROMP) technique.

**Figure 21 jpm-12-00899-f021:**
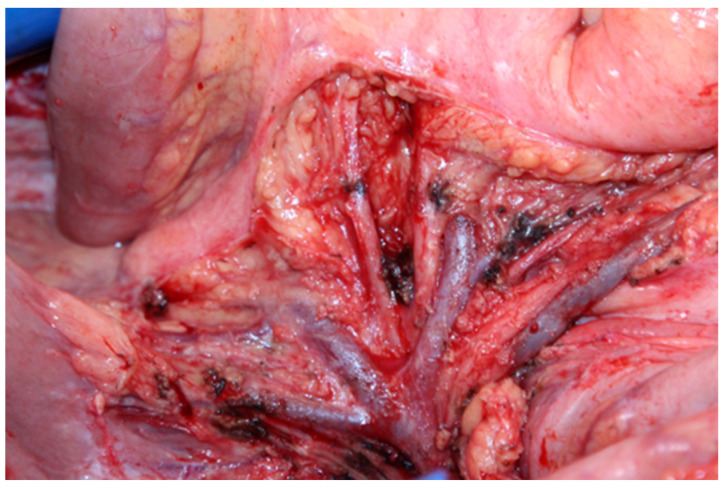
Resecting the peritoneal carcinomatoses from the intestinal mesenterium.

**Figure 22 jpm-12-00899-f022:**
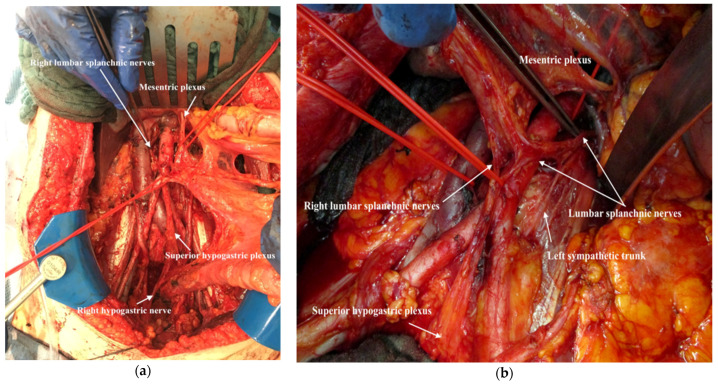
Nerve-sparing systematic para-aortic lymph node dissection as part of the TROMP technique: the right cord of aortic plexus; (**a**) the left cord of aortic plexus (**b**).

**Figure 23 jpm-12-00899-f023:**
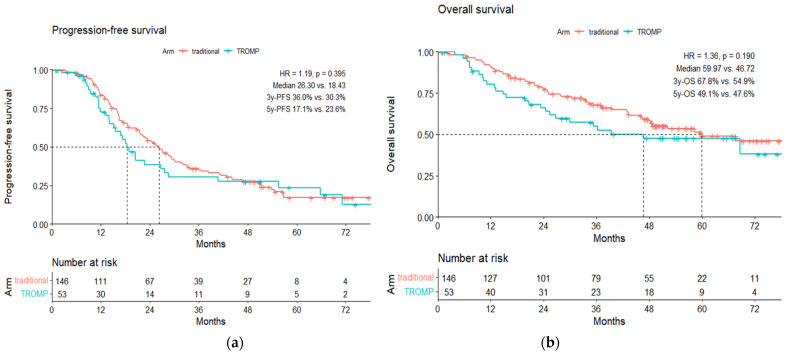
The Kaplan–Meier curves of (**a**) progression-free survival and (**b**) overall survival in both groups.

## Data Availability

The data presented in this study are available on request from the corresponding author. The data are not publicly available due to the privacy and ethical policy of our institution.

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
