# Peer review of "A Promising Approach for Primary Cytoreductive Surgery for Advanced Ovarian Cancer: Survival Outcomes and Step-by-Step Description of Total Retroperitoneal en-Bloc Resection of Multivisceral-Peritoneal Packet (TROMP)"

_jpm, 2022, doi:10.3390/jpm12060899_

Round 1
Reviewer 1 Report
I have read the article with the interest and I would like to congratulate the Authors the development of interesting technique for surgical management of advanced ovarian cancer. In my opinion the study is very attractive, well prepared and contain novel knowledge. However, I would like to recommend minor changes that may improve the manuscript.
1) In my practice, the assessment of tumor resectability in advanced ovarian cancer is performed in 3 steps: 1) physical examination of the patients and evaluation of imaging studies; 2) diagnostic laparoscopy; 3) at the day of actual surgery – by minilaparotomy, where small bowel mesentery and upper abdomen is evaluated. In the case of TROMP technique, the 3rd step of evaluation is abandoned. I would like to ask the Authors to discuss this issue in the Discussion section. I suppose, that TROMP technique is vulnerable for inadequate evaluation of tumor resectability.
2) The Authors use the term “optimal cytoredution” for the cases with no macroscopic residual disease (NGR). In my opinion this is a little bit confusing. I would like to suggest the use of “complete cytoreduction” for NGR, while the term “optimal cytoreduction” should be used for minimal residual disease - tumors less than 1 cm in maximal diameter.
3) The technique is well described, however, there is lack of exact indication when the peritoneal cavity is entered. Is it before 9 step?
4) The Authors try to compare two techniques for surgical treatment of advanced ovarian cancer. Please indicate, whether all of the procedures was performed by the same surgeons? Or what was the percentage of primary Author participation?
5) Please look at the lines 357 – 363 in the Discussion section. In my opinion, these are results and should be presented in the Results section. I would suggest to present to summarize the results in the table.
Author Response
I have read the article with the interest and I would like to congratulate the Authors the development of interesting technique for surgical management of advanced ovarian cancer. In my opinion the study is very attractive, well prepared and contain novel knowledge. However, I would like to recommend minor changes that may improve the manuscript.
Thank you very much for reviewing our manuscript and for the very valuable and encouraging comments.
1) In my practice, the assessment of tumor resectability in advanced ovarian cancer is performed in 3 steps: 1) physical examination of the patients and evaluation of imaging studies; 2) diagnostic laparoscopy; 3) at the day of actual surgery – by minilaparotomy, where small bowel mesentery and upper abdomen is evaluated. In the case of TROMP technique, the 3rd step of evaluation is abandoned. I would like to ask the Authors to discuss this issue in the Discussion section. I suppose, that TROMP technique is vulnerable for inadequate evaluation of tumor resectability.
Thank you very much for the important suggestion.
The physical and radiological pre-operative evaluation is of course an essential part of assessment the tumor resectability. In this study, we did not discuss these points as the concern was on operable patients and not on the pre-operative evaluation. The diagnostic laparoscopy is controversial and we chose in Germany to avoid it because of the following reasons:
- The sensitivity of this tool to predict resectability is in our experience low.
- The risk of laparoscopic site-metastasis is with ca 25% too high.
- The complete resection rate of advanced ovarian cancer is 88% in the TROMP-arm and 100% of patients who were operated to residual tumor less than 10 mm. This very high resection rate warrants us to omit the pre-operative diagnostic laparoscopy.
In surgical step 4, lines 124-128, we mentioned, that we open a peritoneal window in the midline for approximately 3 to 4 cm to take the biopsies and to evaluate the peritoneal cavity. In such cases, we evaluate the infiltration of the small bowel specially the serosa and the infiltration of the hepatoduodenal ligament. These two locations may be a reason for not achieving a complete tumor resection. Therefore, the 3rd step will not be abandoned even in TROMP-Technique.
2) The Authors use the term “optimal cytoredution” for the cases with no macroscopic residual disease (NGR). In my opinion this is a little bit confusing. I would like to suggest the use of “complete cytoreduction” for NGR, while the term “optimal cytoreduction” should be used for minimal residual disease - tumors less than 1 cm in maximal diameter.
We mentioned in lines 90-91: The term optimal cytoreduction was used for cases with no macroscopic residual disease [18], citing the 2010 Gynecologic Cancer InterGroup (GCIG) consensus statement on clinical trials in ovarian cancer: report from the Fourth Ovarian Cancer, which defined the optimal cytoreduction as complete tumor resection
3) The technique is well described, however, there is lack of exact indication when the peritoneal cavity is entered. Is it before 9 step?
The peritoneal cavity is entered in 09 step; the operation will be still performed retroperitoneal.
4) The Authors try to compare two techniques for surgical treatment of advanced ovarian cancer. Please indicate, whether all of the procedures was performed by the same surgeons? Or what was the percentage of primary Author participation?
The first author performed all the TROMP-operations in this study. The conventional arm patients were operated randomly from different gyn oncologists (including the first author) from our center with at least 10-years’ experience.
5) Please look at the lines 357 – 363 in the Discussion section. In my opinion, these are results and should be presented in the Results section. I would suggest to present to summarize the results in the table.
We moved this sentence according to your suggest to the Results section. These results were presented in our previous study und summarized in tables; therefor we did not represented them here again.
Reviewer 2 Report
General comments.
This manuscript presents the survival outcomes and methods of total retroperitoneal en-bloc resection of multi-visceral-peritoneal packages (TROMP).
Basically, this is a well-written paper.
I would recommend it for acceptance after the minor points listed below are addressed.
Specific comments.
Line 102
Please indicate the method used to evaluate the differences in survival. Probably the log-rank test or the Generalized Wilcoxon test.
Line 351
The notation of Fig.22 should be Fig.23.
Line 368-370
If the analysis is feasible, please provide the results of multivariate analysis such as the Cox Regression Analysis with advanced-stage or not, TROMP or not, age, etc. as independent variables. If it is not feasible, please state the reason in the text, for example, the insufficient number of events or the rejection of the proportional hazards assumption, etc.
Author Response
This manuscript presents the survival outcomes and methods of total retroperitoneal en-bloc resection of multi-visceral-peritoneal packages (TROMP).
Basically, this is a well-written paper.
I would recommend it for acceptance after the minor points listed below are addressed.
Thank you very much for reviewing our manuscript and for the very valuable and encouraging comments.
Specific comments.
Line 102
Please indicate the method used to evaluate the differences in survival. Probably the log-rank test or the Generalized Wilcoxon test.
Thank you for the very good hint,we added this sentence to Methodes to clarify this point:
To evaluate the significance of differences in survival, the p-values of the corresponding coefficients of univariable and multivariable Cox regression analysis were used.
Line 351
The notation of Fig.22 should be Fig.23.
Thank you very much for this correction. We changed it accordingly.
Line 368-370
If the analysis is feasible, please provide the results of multivariate analysis such as the Cox Regression Analysis with advanced-stage or not, TROMP or not, age, etc. as independent variables. If it is not feasible, please state the reason in the text, for example, the insufficient number of events or the rejection of the proportional hazards assumption, etc.
The CRA was feasible but did not show any significant difference but for positive lymph node metastases, which is not in line with the results of LION-trial therefore, we decided not to include the CRA in the manuscript.
Cox regression for overall survival
|
term |
HR |
95% CI low |
95% CI high |
p.value |
|
Age |
1.019 |
0.993 |
1.046 |
0.152 |
|
FIGO III |
1.119 |
0.181 |
6.920 |
0.904 |
|
FIGO IV |
0.777 |
0.278 |
2.170 |
0.631 |
|
pT |
0.951 |
0.684 |
1.323 |
0.766 |
|
pN1 |
2.462 |
1.137 |
5.333 |
0.022 |
|
Grading |
1.317 |
0.614 |
2.827 |
0.480 |
Cox regression for progression-free survival
|
term |
HR |
95% CI low |
95% CI high |
p.value |
|
Age |
0.993 |
0.976 |
1.011 |
0.470 |
|
FIGO III |
1.409 |
0.261 |
7.600 |
0.690 |
|
FIGO IV |
0.916 |
0.354 |
2.368 |
0.856 |
|
pT |
1.275 |
0.962 |
1.690 |
0.091 |
|
pN1 |
1.726 |
1.050 |
2.837 |
0.031 |
|
Grading |
1.338 |
0.839 |
2.134 |
0.221 |